# Circulatory Metabolite Ratios as Indicators of Lifestyle Risk Factors Based on a Greek NAFLD Case–Control Study

**DOI:** 10.3390/nu16081235

**Published:** 2024-04-21

**Authors:** Charalambos Fotakis, Athina I. Amanatidou, Maria Kafyra, Vasiliki Andreou, Ioanna Panagiota Kalafati, Maria Zervou, George V. Dedoussis

**Affiliations:** 1Institute of Chemical Biology, National Hellenic Research Foundation, 48 Vas. Constantinou Ave., 11635 Athens, Greece; bfotakis@yahoo.com (C.F.); vicky.x.a@hotmail.com (V.A.); 2Department of Nutrition and Dietetics, School of Health Science and Education, Harokopio University of Athens, 17671 Athens, Greece; aamanatid@gmail.com (A.I.A.); mariakaf@hua.gr (M.K.); nkalafati@gmail.com (I.P.K.)

**Keywords:** MASLD, serum, NMR metabolomics, lifestyle, obesity, metabolite biomarkers

## Abstract

An ensemble of confounding factors, such as an unhealthy diet, obesity, physical inactivity, and smoking, have been linked to a lifestyle that increases one’s susceptibility to chronic diseases and early mortality. The circulatory metabolome may provide a rational means of pinpointing the advent of metabolite variations that reflect an adherence to a lifestyle and are associated with the occurrence of chronic diseases. Data related to four major modifiable lifestyle factors, including adherence to the Mediterranean diet (estimated on MedDietScore), body mass index (BMI), smoking, and physical activity level (PAL), were used to create the lifestyle risk score (LS). The LS was further categorized into four groups, where a higher score group indicates a less healthy lifestyle. Drawing on this, we analyzed 223 NMR serum spectra, 89 MASLD patients and 134 controls; these were coupled to chemometrics to identify “key” features and understand the biological processes involved in specific lifestyles. The unsupervised analysis verified that lifestyle was the factor influencing the samples’ differentiation, while the supervised analysis highlighted metabolic signatures. Τhe metabolic ratios of alanine/formic acid and leucine/formic acid, with AUROC > 0.8, may constitute discriminant indexes of lifestyle. On these grounds, this research contributed to understanding the impact of lifestyle on the circulatory metabolome and highlighted “prudent lifestyle” biomarkers.

## 1. Introduction

Today, the overconsumption of food and the adoption of a sedentary lifestyle in conjunction with excess adiposity result in metabolic variations that may indicate the onset of several disorders, such as abdominal obesity, insulin resistance, dyslipidemia, and an elevated blood pressure. Importantly, lifestyle habits have complicated connections and are often grouped into certain combinations among individuals. In fact, lifestyle risk factors, including an unhealthy diet, physical inactivity, as well as smoking, may have a cumulative impact on health, and have all been linked to an elevated risk of chronic diseases and early mortality [1,2,3,4], as sedulously documented in the literature [5]. Therefore, the early detection and management of chronic diseases is what characterizes preventive medicine, with multiple World Health Organization (WHO) studies deeply oriented towards this concept [4,6,7,8]. It is remarkable that 60% of individuals in the US have one chronic disease and 40% have two or more [9].

Τhe most common chronic liver disease is metabolic dysfunction-associated steatotic liver disease (MASLD) [10]. It refers to a group of diseases spanning from simple hepatic steatosis (SS) or non-alcoholic fatty liver (NAFL) to nonalcoholic steatohepatitis (NASH) and even cirrhosis [11]. Because of the obesity and diabetes epidemics, the prevalence of MASLD is estimated to triple in developed countries and some Asian areas by 2030 [12]. Lifestyle and environmental factors, such as an unhealthy diet, physical inactivity and smoking, as well as metabolic health determinants, including obesity, play an essential role in MASLD development [13,14,15,16].

Metabolic patterns may be used to interpret the adherence to a specific lifestyle by providing insights into metabolic pathways, given the fact that they are easier to correlate with the phenotype, act as direct signatures of biochemical activity, and play a central role in disease development, cellular signaling and physiological control. Other research pertaining to the high-throughput quantification of blood metabolites has elucidated numerous phenotypic aspects (i.e., age, BMI), and even clinical endpoints (i.e., type 2 diabetes, all-cause mortality). Specifically, omics studies have utilized the NMR-based approach to explore the metabolic responses to obesity, dietary exposures [17], physical activity [18], aging, disease onset, mortality and even smoking [19], which constitute the confounding factors of a lifestyle. Therefore, it evident that metabolic profiles are affected by a number of clinical and demographic factors, such as sex, age, body mass index (BMI), smoking, alcohol consumption, and medication use, such as lipid-lowering and anti-inflammatory drugs, with all these parameters constituting a lifestyle.

Few studies to date have provided a measurable fingerprint within the metabolome that determines whether the direction of a lifestyle is curing, or at least modifying, the subject’s metabolome away from or closer to a healthy status. Taking into account the paucity of literature, showing metabolites’ fluctuations in the context of adherence to a specific lifestyle is essential to addressing this issue. It is, therefore, critical to comprehend the frequent occurrence of chronic diseases in people with unhealthy lifestyles and improve public awareness.

On these grounds, this research aspired to investigate whether circulatory metabolomic modifications reflect individual lifestyle patterns and then to pinpoint key “prudent lifestyle” metabolites. For this purpose, we employed data from four modifiable lifestyle risk factors (adherence to Mediterranean diet score, BMI, smoking, Physical Activity Level—PAL) to create a lifestyle risk score (LS). Our cohort included 89 MASLD patients and 134 controls who were assessed for their dietary patterns in our previous study [19,20]. The study population was categorized into four groups based on the LS score, where a higher score group is indicative of a less healthy lifestyle. Furthermore, the serum NMR-based metabolomics data of the cohort were investigated through exploratory analysis and biomarker and pathway analysis in an effort to detect and quantify the cohort’s metabolic response to the impact of these four modifiable lifestyle risk factors, as expressed by the lifestyle classifications, and further investigate their involvement in the incident of chronic diseases.

## 2. Materials and Methods

### 2.1. Study Population and Design

This study used a sample set of 223 participants (89 MASLD patients and 134 controls) from a Greek case–control study [20]. Information regarding the study’s methodology is available elsewhere [20]. Adults without self-declared concurrent liver damage were tested for MASLD at the time of enrollment. Any congenital or acquired liver disease, chronic viral hepatitis, hepatotoxic drug exposure, excessive alcohol consumption, life-threatening diseases or psychiatric disorders impairing the patient’s ability to provide written informed consent, and pregnancy or lactation were all exclusion criteria. All research participants were briefed about the study’s objectives and completed a written consent form. This study was approved by the Ethics Committee of Harokopio University of Athens (38074/13-07-2012), based on the Helsinki Declaration.

#### MASLD Diagnosis

Herein, liver ultrasound (U/S) was used as the imaging method for disease diagnosis. Liver biopsy remains the diagnostic reference standard for MASLD, due to its high accuracy in detection and staging of the disease [21]. However, biopsy has some disadvantages; it is invasive, expensive, and can occasionally result in serious side effects. On the other hand, U/S is recommended as the first-line examination for individuals with a high risk of this disease, given that it is reasonably priced, widely accessible, non-invasive, suitable for follow-up exams, and incredibly practical and friendly to the patients. Thus, liver U/S was applied in all participants to assess the stage of MASLD. Depending on the results, the participants were then divided into cases and controls. Controls were those with no (healthy liver subjects) or mild hepatic steatosis (grade 1 subjects), and cases were those with moderate (grade 2 subjects) and severe hepatic steatosis (grade 3 subjects).

### 2.2. Data Collection

Demographic, family, as well as individual medical history interviews were undertaken by trained personnel. Anthropometric measures were also obtained for all participants. Body weight, height, and waist circumference were all measured, and the BMI was calculated by dividing weight (kg) by height (m^2^). The waist and hip circumferences were obtained, and the waist to hip ratio (WHR) was calculated. Based on their smoking status, individuals were categorized as current smokers or non-smokers. The validated brief self-reported Athens Physical Activity Questionnaire (APAQ) [21] was used to collect physical activity data, which were then utilized to compute each participant’s PAL and total energy expenditure. A semi-quantitative self-reported food frequency questionnaire (FFQ) [22] was used to quantify dietary habits. After a 12 h overnight fast, blood tests were performed; these evaluated the lipidemic and glycemic profile, as well as liver enzymes. The Friedewald equation was used to calculate the low-density lipoprotein cholesterol (LDL-C) and the homeostatic model assessment (HOMA-IR) was used to quantify the degree of insulin resistance.

### 2.3. Construction of Lifestyle Risk Score

The LS was constructed using data from four modifiable lifestyle factors, specifically adherence to the Mediterranean diet, as evaluated by using the MedDietScore, BMI, smoking and PAL. The MedDietScore [23] was calculated using the following nine dietary components: non-refined cereals, potatoes, fruits, vegetables, legumes, fish, red meat and products, poultry, and full-fat dairy products. A MedDietScore ≤ 20 (median) was considered to reflect an unhealthy diet. The high-risk group for smoking (current smokers vs. non-smokers) included those with a current smoking status, for BMI, those with a BMI ≥ 24.99 kg/m^2^ and for PAL, those with PAL ≤ 1.37 (median). Participants received a score of 2 for each of the four aforementioned lifestyle factors if they engaged in an unhealthy lifestyle; otherwise, they received a score of 1. The sum of these four scores resulted in a total lifestyle risk score ranging from 3 to 8. Higher scores imply an unhealthier way of life. Four groups were created based on the overall LS: LS 3–4 (score 3–4), LS 5 (score 5), LS 6 (score 6), LS 7–8 (score 7–8).

### 2.4. Univariate Analysis of the Anthropometrics, Physiological Parameters and LS Score

Categorical variables are given as absolute frequencies, and quantitative variables are shown as mean ± SD, since all of them were considered as normally distributed based on the central limit theorem (CLT). For the evaluation of differences between the LS groups, analysis of variance (ANOVA) with Tukey’s post hoc test was implemented. The Chi-square test was applied for the comparison of the LS groups of categorical variables represented as numbers.

### 2.5. Circulatory Metabolome Analysis

#### NMR-Metabolomics Pipeline

Serum NMR spectra were acquired using a Varian-600 MHz NMR spectrometer equipped with a ^1^H{^13^C/^15^N} 5 mm PFG Automatable Triple Resonance probe at 25 °C, as previously described [24]. Briefly, the CPMG pulse sequence with presaturation water suppression was applied, collecting 128 transients with 64 K data points, with a relaxation delay of 5 s and an acquisition time of 4 s. Details of the serum sample pretreatment, including methanol (1:2 *v*/*v*) extraction, centrifuging, as well as reconstitution in buffer, are also presented in [24].

The ^1^H-NMR spectra process and metabolite annotation are also described in [24]. Briefly, MestreNova (v. 10.1) software was applied for the preprocess of the spectra (manual phase correction, automatic baseline correction and sinc apodization), normalization to the total area, binning with 0.001 ppm, and peak alignment against the superimposed spectrum.

The application of an in-house automated metabolite identification platform, i.e., Metaboneer [25], enabled the identification of 42 metabolites.

### 2.6. Multivariate Data Analysis

#### 2.6.1. Post Processing of NMR Spectral Data

SIMCA-P (version 14.0, Umetrics, Umeå, Sweden) was applied. The spectral data were mean-centered and Pareto scaled (Par), and the unsupervised Principal Component Analysis (PCA) models were extracted at a confidence level of 95%. The mathematical background and applications of these methods have been extensively discussed.

#### 2.6.2. Identification of Important Features

First, principal component analysis was employed in order to visualize any relations (trends, outliers) among the observations (samples). A PCA model estimates the systematic variation in a data matrix using a low-dimensional model plane. The spectral data were mean-centered with Pareto scaling (Par) and the PCA model was extracted at a confidence level of 95%.

Loading and contribution plots were extracted to reveal the variables that bear class discriminating power.

#### 2.6.3. Model Validation

The quality of the models (PCA) was described by the goodness-of-fit R^2^ (0 ≤ R^2^ ≤ 1) and the predictive ability Q^2^ (0 ≤ Q^2^ ≤ 1) values. The R^2^ explained the variation, thus constituting a quantitative measure of how well the data of the training set were mathematically reproduced. The overall predictive ability of the model was assessed by using the cumulative Q^2^, representing the fraction of the variation of Y that could be predicted by the model, which was extracted according to the internal cross-validation default method of the software SIMCA-P 14. Q^2^ is considered a de facto default diagnostic parameter for validating models in metabolomics. In particular, the difference between the goodness of fit and the predictive ability remained always lower than 0.3 (R^2^X(cum) − Q^2^(cum) < 0.3), and the goodness of fit never equaled one (R^2^X(cum) ≠ 1). Therefore, if the extracted models abided by these rules, their robustness and predictive response were enhanced and over-fitting was effaced.

#### 2.6.4. Metabolic Markers and Associated Metabolic Pathways

The web-based MetaboAnalyst (V5.0) platform (https://www.metaboanalyst.ca/ (accessed on 1 March 2023)) was utilized for biomarker discovery, classification and the pathway mapping of metabolites exhibiting AUROCs > 0.7 to enable the exploration of disease-related metabolites and pinpoint the most relevant pathways.

The ASCII file containing the aligned spectra after their reduction into spectral buckets of 0.001 ppm was used as the input data type for analysis with MetaboAnalyst. From the available modules, we also used Biomarker Analysis to extract ROC curves, and the levels of metabolites exhibiting a sub-optimal and higher performance were framed in box plots.

The metabolites exhibiting superior AUROC accuracy were compiled in a one-column compound list that was in turn implemented for Enrichment Analysis and Pathway Analysis. Specifically, the metabolic pathway analysis (MetPA) algorithms included the hypergeometric test for over-representation analysis, and the relative betweenness centrality for pathway topology analysis based on the KEGG library was applied. Metabolic pathways with a hypergeometric test *p*-value less than 0.05 were considered to be disturbed. Metabolite Set Enrichment Analysis (MSEA) was also performed for the metabolites exhibiting AUROC  >  0.8, and a metabolite set library based on the disease signatures’ library for the blood substrate was applied. MSEA monitors whether these metabolites are represented more often than expected in an attempt to identify biologically meaningful patterns.

## 3. Results and Discussion

### 3.1. Participant Characteristics

In Table 1, the main characteristics of the study participants, categorized based on the LS group, are shown. This study comprised 223 participants, 15 of whom had LS 3–4, 50 of whom had LS 5, 82 of whom had LS 6, and 76 of whom had LS 7–8.

The number of MASLD patients was observed to increase progressively with increasing LS. The mean LS of cases was also higher than that of the controls (*p*-value < 0.0001) (Figure 1). Sex differences were also detected between the LS groups. A higher FLI, BMI and WHR, lower MedDietScore, lower physical activity, higher current smoking status, lower AST/ALT ratio, higher insulin and HOMA-IR, lower high-density lipoprotein (HDL) and higher triglyceride (TG) content were found to characterize the individuals with increased LS (*p*-value < 0.05). The incidence of hypertension and metabolic syndrome (MetS) rose progressively with increasing LS. As shown in Table 1, the LS 7–8 group is the most burdened compared to the others.

### 3.2. The NMR Metabolic Patterns Interpret Lifestyle Trends

The circulatory metabolites’ composition when facilitating multivariate data analysis may enable the discerning of metabolic variations and their relation to variables of interest, such as nutritional habits, smoking, obesity and physical exercise, thus representing the adherence to a specific lifestyle.

#### 3.2.1. Exploratory Analysis

A PCA model with two components (A = 2) on the host of 223 serum NMR spectra provided an overview of the samples’ clustering, highlighted outliers and enabled an evaluation of whether a differentiation could be observed (Figure 2). Interestingly, the unsupervised analysis revealed a clear linear trend along the second component, essentially portraying the transition from a sedentary to a unhealthy lifestyle (1st and 2nd quadrants) to an active and healthier lifestyle (3rd and 4th quadrants). In particular, the PCA model pinpoints a clear grouping of subjects categorized as LS 3–4 in the 3rd and 4th quadrants; meanwhile, on top of this group lie the subjects categorized as LS 5, across the second component (PC2) are located the subjects categorized as LS 6, then passing to the 1st and 2nd quadrants are the subjects categorized as LS 7–8. This result further enhances the notion that lifestyle modulates, in a significant way, metabolic profiles and is attributed to an ensemble of factors, as represented by the lifestyle categorization.

More information from the PCA model (Figure 2) can be obtained by extracting the contribution plots for each lifestyle group. For instance, a contribution plot for the samples from LS 3–4 was extracted (Figure 3A) by comparing their average values to the average value of the rest of the samples in order to pinpoint the variables that contribute to the clustering of each lifestyle group, thus showing the metabolic responses that are caused by adherence to a lifestyle. Specifically, the up-regulation of formic acid, the glutamic acid of the proteinogenic amino acid glycine, the phospholipid constituent phosphorylcholine (ChoP), glycine, as well as the biogenic amine creatine was observed in samples from LS 3–4, while BCAAs (valine, leucine, isoleucine) and methionine displayed a decreased concentration.

The metabolite phosphorylcholine (ChoP) is evidence of fruit and vegetable intake [26]. ChoP is the hydrophilic polar head group of some phospholipids that also consist, in part, of the potent inflammation mediators PAF (platelet-activating factor) and PAF-like lipids. In fact, ChoP has various properties that could potentially promote and protect against disease, depending on the pathogen and the type of inflammatory reaction [27].

Formic acid plays a central role in human metabolism, contributing to nucleotide synthesis, while alterations in formate metabolism have been related to human pathological conditions such as cancer, neurological disorders, obesity and CVD. Higher levels of formic acid have been framed as a metabolic response to a lifestyle involving weight loss treatment [26,28].

Glycine constitutes a potent antioxidant-scavenging free radical that is fundamental to the antioxidative defense of leukocytes. Additionally, this metabolite bears anti-inflammatory, immunomodulatory, and cytoprotective attributes. In one study, glycine was positively attributed to an adherence to a healthy lifestyle [29]. Hasegawa et al. [30] has studied the inverse relationship between glycine and the occurrence of MASLD. Congruent with this observation, the high concentration of glycine was monitored in LS 3–4, the group in which no sample was diagnosed with MASLD.

Glutamic acid is reported to be an ample transmitter in the nervous system, promoting 40% of all synapses in the brain. This metabolite facilitates the transport of reducing agents across the mitochondrial membrane and regulates glycolysis and the cellular redox state through the malate/aspartate shuttle [31]. Other research has documented the impact of glutamate imbalance on glutamatergic neurotransmission through anxiety and stress [32,33]. Another study estimated a healthy lifestyle index (encompassing parameters such as diet, BMI, physical activity, lifetime alcohol, smoking, diabetes, hepatitis) that can be attributed to the concentration of serum metabolites such as glutamic acid and phosphatidylcholine [34].

The endogenous metabolite creatine is deemed essential in the network of energy transfer by cardiovascular research [35]. In our study, increased levels of creatine were attributed to the LS 3–4 group, which follows an active and healthier lifestyle. In alignment with our results, the levels of creatine have previously been related to an improved plasma metabolic profile in women [28]. There is substantial evidence that this metabolite may prevent the occurrence of fatty liver in high-fat and choline-deficient diets, as well as in hepatoma cells in vitro [36].

The low concentration of BCAAs may be the result of a decreased consumption of red and processed meat. Consistent evidence documents that the increased consumption of red meat and processed meat can be attributed to an increased risk of diabetes and cardiovascular disease (CVD) [37]. BCAAs have been related to the risk of diabetes or CVD in previous studies [38]. The consumption of carbonated drinks and even fruit juice has also been attributed to increased leucine and isoleucine levels in the plasma substrate [39].

In Figure 3B, the class discriminant spectral regions for LS 5 are displayed, thus indicating increased contents of glutamine and arginine and low concentrations of BCAAs (Valine, leucine, isoleucine) and methionine. In fact, the amino acid arginine acts as a precursor for the synthesis of protein, nitric oxide, creatine, polyamines, agmatine, and urea. It is also well known that among the amino acids, arginine is a potent activator of mammalian target of rapamycin complex 1 (mTORC1), a metabolic rheostat and central signaling hub that, in accordance with nutrient availability, determines anabolic and catabolic processes [40].

Subsequently, the contribution plot of LS 6 (Figure 3C) showed a high content of glycine and a low concentration of BCAAs (valine, leucine, isoleucine), glucose and tyrosine. The low concentration of glucose may be attributed to a reduced total energy intake [41,42].

Finally, the contribution plot of LS 7–8 (Figure 3D) indicated an extremely high content of BCAAs (valine, leucine, isoleucine), a high content of acetoacetate, methionine, tyrosine and creatinine, and low concentrations of 3-hydroxy butyrate, glutamine, choline and phosphorylcholine. This metabolic profile is an inverse equivalent of the metabolites that characterize LS 3–4. This further proves that these two groups have different lifestyle habits. Several research investigations have found higher plasma BCAA levels in MASLD patients [43,44,45]. Interestingly, the LS 3–4 group, which does not include subjects with MASLD, displayed downregulated levels of BCAAs, whereas the BCAA levels were greater in the LS 7–8 group, which encompassed patients with severe liver disease [43,44,45]. Lake et al. [46] discovered that the serum leucine, isoleucine, and valine levels, which comprise BCAAs, increased significantly as steatosis proceeded to NASH. This increase has been associated with hepatic fat accumulation in the early stages of MASLD. The tyrosine levels were positively related to NAFLD severity for both males and females in our recent study [24], as also observed in other studies [43,47,48,49,50,51]. Another study on the serum metabolome highlighted that elevated levels of BCAAs and aromatic amino acids are predictive biomarkers of T2D [52].

Another metabolite that is increased in LS 7–8 is creatinine, an endogenous substrate stemming mainly from creatine in muscle and related to muscular function [53]. In addition, the creatinine concentration in blood and urine serves as a marker of kidney disease. The ATTICA study revealed that greater adherence to the Mediterranean diet is associated with lower serum urea and creatinine levels [54]. Interestingly, recent studies have attributed the serum uric acid/creatinine ratio to NAFLD severity [55].

Fluctuations in the ketone bodies (acetoacetate, 3-hydroxybutyrate) can be associated with dietary habits that involve the consumption of a carbohydrate-restricted, high-fat diet. Another factor that affects the levels of 3-hydroxybutyrate in the blood is the occurrence of ketosis. Acetoacetate (AcAc) constitutes a ketone body primarily produced in the liver when the conditions of excessive fatty acid breakdown occur, such as in diabetes mellitus resulting in diabetic ketoacidosis. This metabolite is an indispensable energy source in times of limited glucose supply, like bHB; acetoacetate synthesis is higher when undergoing fasting, endurance exercise, and malnutrition. An increased concentration of such ketone bodies may indicate diabetic hyperglycemia and also point to a disturbed glucose metabolism in the prediabetic state [56].

#### 3.2.2. Distinct Metabolite Markers

Furthermore, we performed supervised analysis to validate the distinct metabolomic bouquet and to extract potential metabolites that could serve as biomarkers. An OPLS-DA model (Figure 4) was extracted by comparing the healthiest expected lifestyle (3–4) to the unhealthiest lifestyle (7–8).

The OPLS-DA discrimination was validated through permutation testing and receiver operator characteristic (ROC) curves (Appendix A).

The OPLS-DA model resolved the metabolic variation, and two clusters (lifestyle 3–4 and 7–8) were evident along the first component (Figure 4). The derived S-line plot pinpointed the glucogenic amino acids alanine and isoleucine, acetoacetate, methionine, acetic acid, serine, and threonine as metabolite markers positively correlated to lifestyle 7–8 (Figure 4b). On the other hand, the samples corresponding to lifestyle 3–4 exhibited elevated levels of 3-hydroxybutyrate, glutamic acid, phosphorylcholine, betaine, glycine and formic acid.

#### 3.2.3. Receiver Operating Characteristic (ROC) Curve Analysis for Metabolite Markers

The calculation of the AUC enabled us to further assess a quantitative measure for the discriminatory potential of the OPLS-DA model (Figure 4) and avoid false selection.

Biomarker analysis (Figure 5) confirmed the ability of the proposed OPLS-DA to discriminate biomarkers between the two groups, thus revealing a high Area Under the Receiver Operating Characteristic curve (AUROC).

Specifically, the ROC curves for each of the discriminant metabolites were estimated by the use of MetaboAnalyst, as well as the top-ranked metabolite concentration ratios (based on *p*-values). The latter choice may bear more information than the two corresponding metabolite concentrations alone.

In fact, the diagnostic accuracy exhibited by leucine/serine, formic acid, formic acid/tyrosine, formic acid/dimethylamine, alanine/methylamine, alanine/serine, alanine/ethanolamine, formic acid/threonine, leucine, acetic acid, leucine/phosphorylcholine, formic acid/acetoacetate, and leucine/serine was fair (0.8 > AUROC > 0.7) (Appendix A), while an optimal diagnostic accuracy (AUROC > 0.8) was exhibited by leucine/formic acid, formic acid/L-tyrosine, formic acid and alanine/formic acid.

The metabolite ratios of leucine/formic acid and alanine/formic acid increased significantly in the samples from lifestyle 7–8, whereas formic acid/L-tyrosine and formic acid decreased significantly compared to the healthy samples from lifestyle 3–4 (Figure 5).

These metabolite ratios enable biomarker candidates to determine adherence to a healthy or unhealthy lifestyle with increased accuracy. The proposed biomarkers may not be regarded as stand-alone lifestyle biomarkers, even if they could bring additional value to early unhealthy lifestyle prediction when included in multi-biomarker approaches. In particular, these ratios may constitute a metabolic trigger for the onset of chronic diseases, but such a supposition needs further validation in independent studies.

#### 3.2.4. Metabolite Pathway Analysis

The identification of metabolic signatures related to lifestyle has garnered scientific interest, but in order to gain meaningful information and design appropriate preventive interventions, we must determine the metabolic pathways that are involved in the development of the disease.

With the use of MetaboAnalyst 5.0 [2], we performed metabolite pathway analysis to determine relevant metabolic pathways based on the identified metabolites with AUROCs > 0.8 (i.e., L-alanine, L-leucine, L-tyrosine, formic acid). The pathway analysis results are displayed in Figure 6. This reveals that lifestyle elicits significant changes in the circulating metabolome, reflecting alterations in several metabolic pathways.

Specifically, in response to adherence to a specific lifestyle, the primary disturbed statistically significant pathways (*p* < 0.05) containing at least two compounds include Aminoacyl-tRNA biosynthesis; phenylalanine, tyrosine and tryptophan biosynthesis; valine, leucine and isoleucine biosynthesis; ubiquinone and other terpenoid–quinone biosynthesis; and phenylalanine metabolism (Appendix A). This reveals mainly a perturbed amino acid metabolism.

Perturbations in metabolite levels usually occur in a dysregulated or exacerbated state of a biological system, i.e., low or extreme physical activity [57], affecting the metabolism of nitrogenous substances. A study by Lu et al. [58] also documented the pathways that refer to amino acid imbalance when subjects adhere to multiple healthy lifestyle factors and when the circulatory metabolome is affected. Amino acids are both the dynamic structural building blocks of proteins and moreover are active signaling molecules that regulate metabolism.

Furthermore, the crosstalk between the metabolic pathway of ubiquinone, terpenoid quinone biosynthesis and CAD has also been highlighted, proposing markers for diagnosis as well as detection using the subjects’ serum [59].

These findings facilitate the expansion of biomarker research in the perturbed metabolic pathways.

#### 3.2.5. Metabolite Enrichment Analysis

We performed a hypergeometric test using over-representation analysis and pathway topology analysis (Figure 7).

The metabolic markers with AUROC > 0.8 were incorporated into the metabolite enrichment analysis, in order to investigate metabolite attributions and uncover the disease signatures in our serum samples by highlighting the probable association of lifestyle with other medical conditions (Appendix A). The metabolic profiles were associated with possible disease endpoints, including types of seizures, heart failure, and mental disorders.

In particular, the enrichment analysis pointed towards a link between the adherence to a lifestyle and the risk of heart failure, myocardial injury, inflammatory diseases and seizure disorders. This finding is in alignment with recent literature suggesting that lifestyle might constitute a risk factor for chronic disease, despite the fact that the intrinsic causality is still under examination [60].

To date, there is accumulating evidence on the impact of weight management, exercise, nutrition and dietary composition on cardiovascular disease (CVD) [61]. In agreement with this, our results also found that heart failure and myocardial injury are a possible outcome of an unhealthy lifestyle.

The low overall volume of habitual PA/exercise in LS 7–8 is positively correlated with metabolomic signatures that align with worse cardiometabolic health, while physically active subjects exhibit a “coherently healthier metabolic profile compared to their inactive counterparts”. This may aid in reducing the increased cardiometabolic risk attributed to a sedentary/inactive lifestyle [62].

Despite the beneficial health effects of physical activity being well recognized, physical inactivity is estimated to amount to 9% of premature mortality worldwide [3]. Over the last 30 years, overweight and obesity have resulted in a concomitant increase in the prevalence of co-morbidities such as cardiovascular disease and type 2 diabetes. Sedentary behavior has also been framed as a factor that impacts this epidemic and is related to a high risk of all-cause mortality [63].

Sedentariness is distinct from physical inactivity, as it involves certain activities in a reclining, seated, or lying position, requiring very low energy expenditure. The issues with this behavior have been associated with a lack of movement, and also with the stimulation provided by replacing these activities. It has been proposed as an independent predictor of metabolic risk even when a subject abides by current physical activity guidelines. In recent decades, changes in the activity profile of subjects have marked the replacement of vigorous physical activity and sleep with cognitive work. This is a potential neurogenic stress factor, taking into account its hormonal and neurophysiological health impact.

Lifestyle choices may also impact the occurrence of seizures, with certain seizure triggers already associated with an unhealthy lifestyle, such as a lack of sleep, overexertion or physical fatigue, physical or emotional stress, alcohol and other drug use [64].

Interestingly, LS 7–8 was associated with the lowest physical activity and highest obesity rates. Its metabolic profile was positively correlated with BCAA. These are significant metabolites pertaining to numerous complex metabolic pathways, such as neurotransmission, and disorders of the amino acid metabolism that may impact neurological functions in humans. Circulating BCAAs may be introduced to the brain to enable glutamate biosynthesis and may either impede or evoke acute seizures [65]. This strengthens the case that the metabolomic response is related to a person’s physical activity [57].

The probable effect that amino acid metabolism has on triggering heart diseases can be explained, since most amino acid catabolic activities are found in the liver. Branched-chain amino acid (BCAA) catabolism demands activity in several non-hepatic tissues, such as cardiac muscle, the diaphragm, brain and kidney. This may in part explain the development of MASLD in LS 6 and LS 7–8. Liver damage and the relationship between MASLD and cardiovascular disease (CD) tends to be the primary driver of mortality in these individuals.

Other studies support the case that chronic epilepsy may negatively influence the structural integrity of the heart and its vasculature, in turn causing “The Epileptic Heart”. These negative influences result in cardiac electrical instability, susceptibility to arrhythmias, and even variations in autonomic function [66].

Phenylketonuria was also revealed as a potential disease, but since it is an inherited metabolic disorder that impairs the phenylalanine (Phe) metabolism, it is probably loosely connected to lifestyle [67]. However, it may appear in a person with PKU after the consumption of protein-rich foods, or grains or aspartame (milk, cheese, nuts, meat, bread, pasta).

These findings in general confirm a bouquet of disease-related metabolites and promote a unified metabolomic background for common diseases. This potential clinical utility for NMR-based metabolomics may translate into another source of discriminatory information to assess the risk of a number of diseases.

## 4. Conclusions

The present study investigated circulatory metabolomic modifications in relation to lifestyle and thus to the covariates that influence it, such as BMI, dietary habits, physical exercise and smoking. In fact, alterations in the serum metabolite concentration can significantly influence cellular and whole-body function, resulting in acute and chronic human diseases. Since metabolite fluctuations constitute the basis of most human diseases, approaches that modulate the metabolism must constitute the “vanguard” of therapeutic and preventative steps. Risk stratification is important in order to identify, as soon as possible, high-risk individuals and establish disease prevention. Therefore, the rapid monitoring of the circulatory metabolome that reflects individual lifestyle patterns may play a preventive role in the development of chronic disorders.

Our results help to understand the influence of lifestyle on the circulatory metabolome and pinpoint key “prudent lifestyle” metabolites. Metabolite–disease associations have also been assessed, providing complementary information that aids in the prediction of disease risk. The validation steps found that tyrosine, formic acid, leucine and alanine are the putative metabolite markers that bear the impact of lifestyle in the serum. This may enable correlations to be established between blood metabolites and the confounding factors contributing to the overall biochemical picture of lifestyle. The potential connections of liver diseases with the adherence to a lifestyle is documented by a number of studies and should be considered in clinical practice. Last but not least, our findings implicate that the metabolic response to a specific lifestyle could potentially trigger the development of MASLD, heart issues and seizures.

## Figures and Tables

**Figure 1 nutrients-16-01235-f001:**
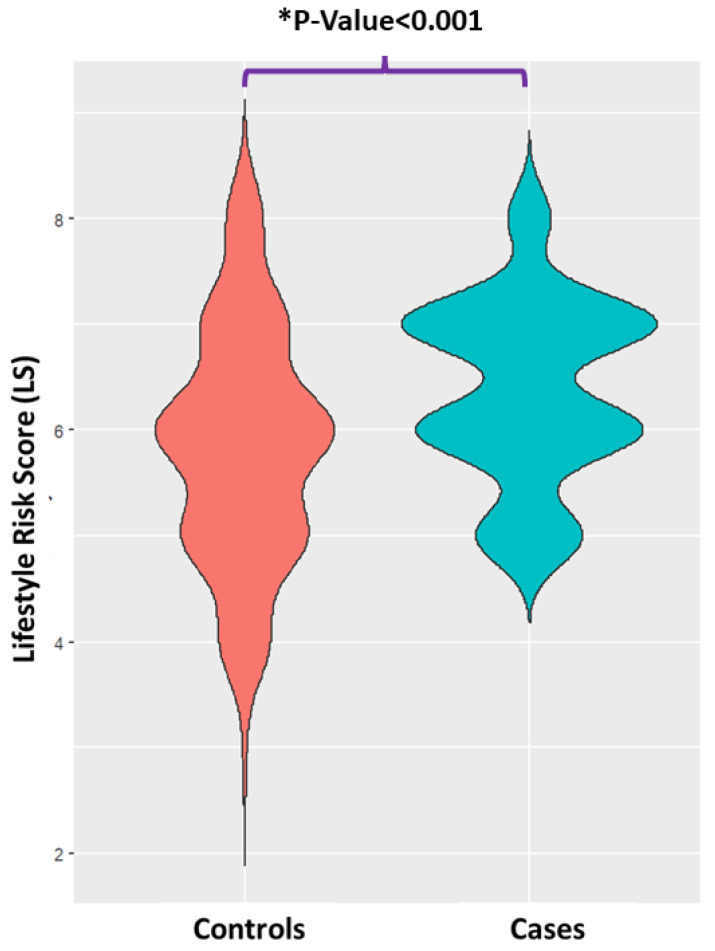
The violin plot depicts the distribution of the lifestyle risk score (LS) for MASLD cases and controls. Notes: *p*-value was obtained using independent samples *t*-test. * Statistically significant (*p*-value < 0.05) between groups.

**Figure 2 nutrients-16-01235-f002:**
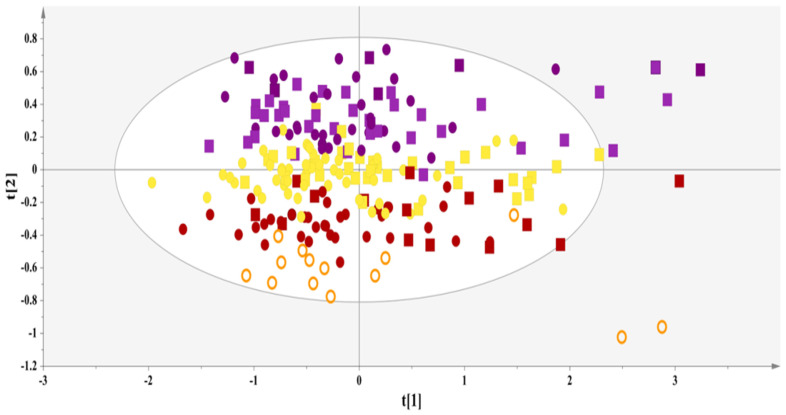
A PCA model: A = 2, N = 223; R^2^X(cum) = 0.48, and Q^2^(cum) = 0.39. (circles = control, square = MASLD, orange = lifestyle group 3–4, red = lifestyle group 5, yellow = lifestyle group 6, purple = lifestyle group 7–8).

**Figure 3 nutrients-16-01235-f003:**
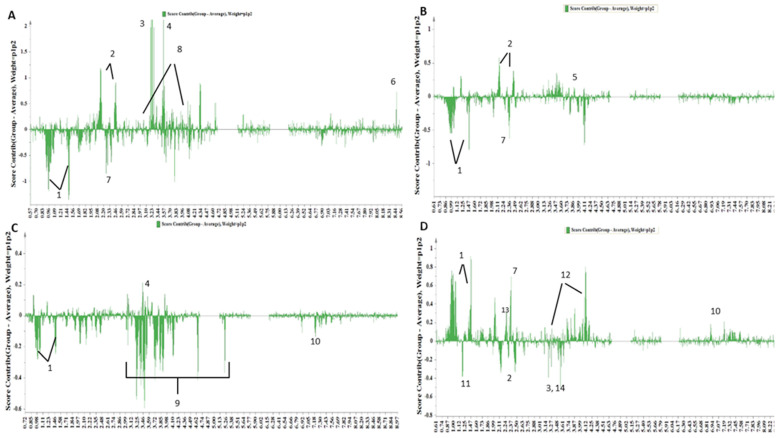
PCA contribution plots of the samples of LS 3–4 (**A**), 5 (**B**), 6 (**C**), 7–8 (**D**) (1: valine, leucine, isoleucine, alanine, 2: glutamic acid, 3: phosphorylcholine, 4: glycine, 5: arginine, 6: Formic acid, 7: methionine, 8: creatine, 9: glucose, 10: tyrosine, 11: 3-hydroxybutyric acid, 12: creatinine, 13: acetoacetate, 14: choline).

**Figure 4 nutrients-16-01235-f004:**
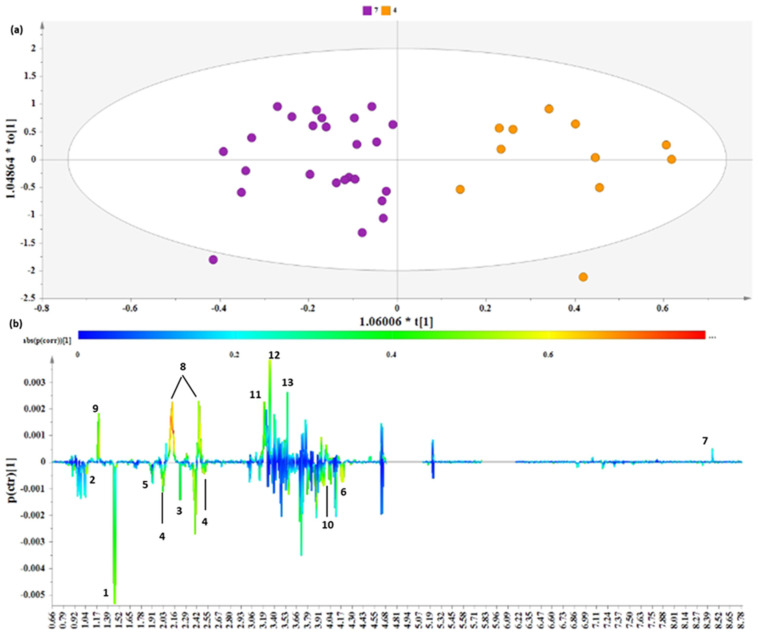
Comparison samples from lifestyle groups 3–4 and 7–8. (**a**) OPLS–DA model; A = 1 + 1, N = 80; R^2^X(cum) = 0.38, R^2^Y(cum) = 0.75 and Q^2^(cum) = 0.46, *p*-value = 1.29208 × 10^−9^. Purple circles = samples of lifestyle 7–8, orange circles = samples of lifestyle 3–4. (**b**) S–line plot (1. alanine, 2. isoleucine, 3. acetoacetate, 4. methionine, 5. acetic acid, 6. threonine, 7. formic acid, 8. glutamic acid, 9. 3-hydroxybutyrate, 10. serine, 11. phosphorylcholine, 12. betaine, 13. glycine).

**Figure 5 nutrients-16-01235-f005:**
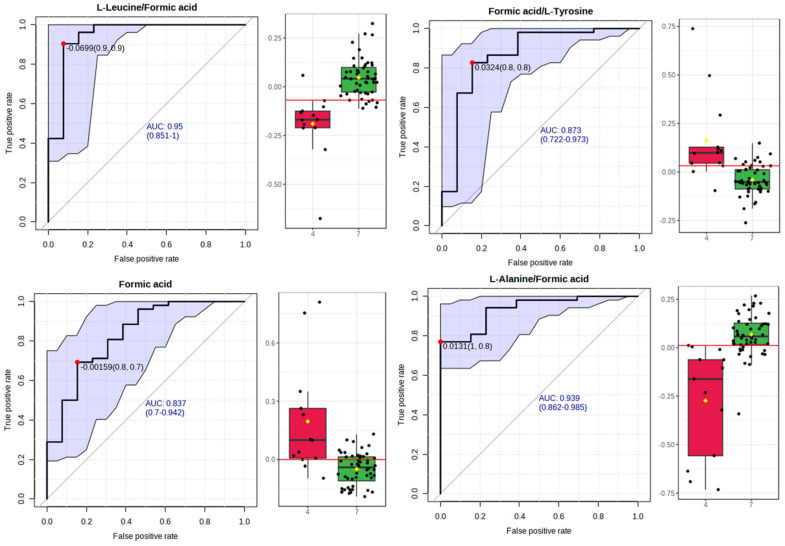
Box plots and ROC curves for each metabolite differentially abundant between lifestyle 3–4 (red bar graphs) and 7–8 (green bar graphs).

**Figure 6 nutrients-16-01235-f006:**
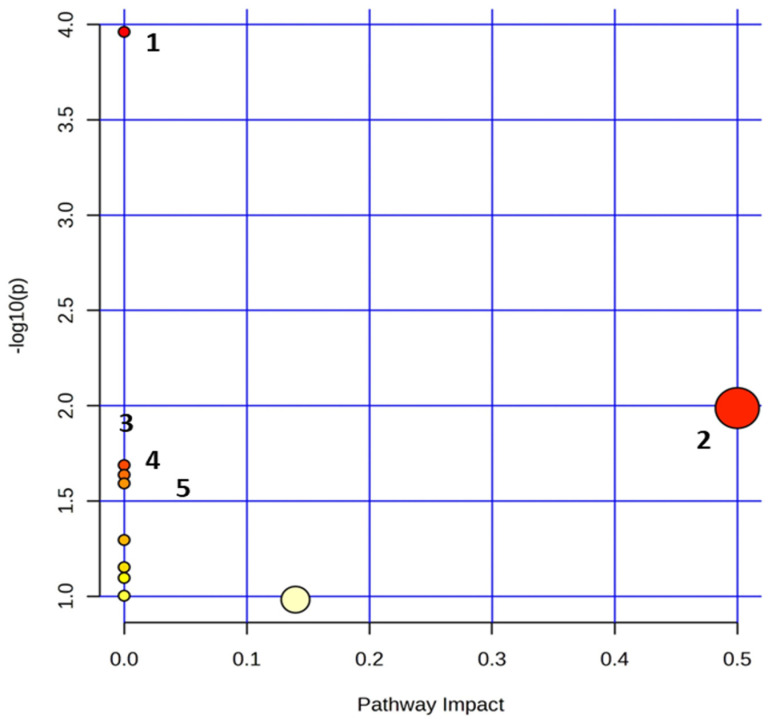
*Summary plots for over-representation analysis of the serum substrates*. 1. Aminoacyl-tRNA biosynthesis, 2. phenylalanine, tyrosine and tryptophan biosynthesis, 3. valine, leucine and isoleucine biosynthesis, 4. ubiquinone and other terpenoid–quinone biosynthesis, and 5. phenylalanine metabolism. The size of the circle varies accordingly to the higher centrality of the metabolite in the related pathways (impact value).

**Figure 7 nutrients-16-01235-f007:**
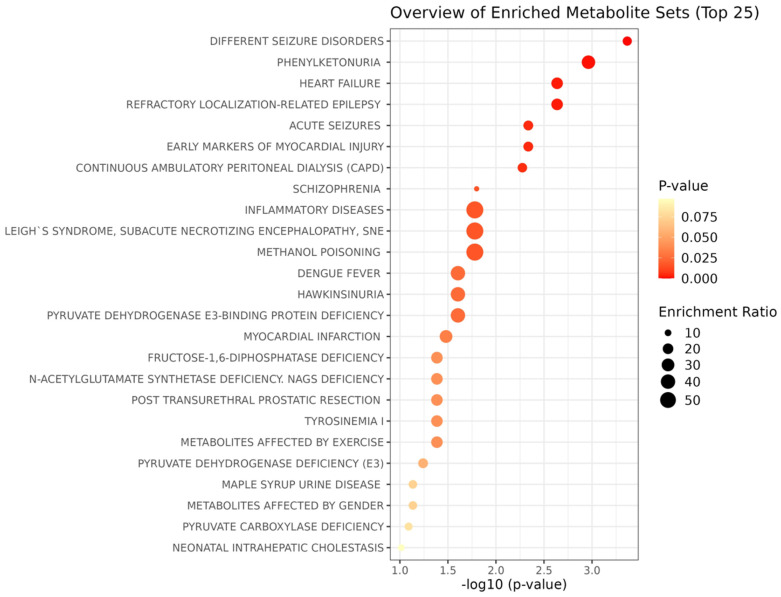
Enrichment analysis results based on the serum signature disease.

**Table 1 nutrients-16-01235-t001:** Baseline characteristics of the participants according to the lifestyle risk score (LS).

	LS 3–4 (N = 15)	LS 5 (N = 50)	LS 6 (N = 82)	LS 7–8 (N = 76)	*p*-Value
**Cases/Controls** **(89/134)**	0/15	15/35	32/50	42/34	**2.15 × 10^−4^**
**FLI**	8.34 ± 7.36 **** † ¶**	30.90 ± 29.46 ****^,^***	38.2 ± 30.94 **† #**	51.90 ± 28.38 **¶ * #**	**3.24 × 10^−7^**
**Age (years)**	43.85 ± 11.27	48 ± 10.75	46.20 ± 12.08	44.93 ± 12.13	0.413
**Sex** **(Females—%)**	73.3	62	65.8	46	**0.04**
**MedDietScore**	23.23 ± 1.54 **† ¶**	21.80 ± 2.42 **§ ***	19.41 ± 3.84 **† § #**	17.46 ± 4.743 **¶ * #**	**4.48 × 10^−10^**
**PAL**	1.51 ± 0.10 **¶**	1.52 ± 0.21 *****	1.45 ± 0.24 **#**	1.27 ± 0.17 **¶ * #**	**2.01 × 10^−10^**
**Smoking status** **(current smoker—%)**	0	8	30.4	55.2	**1.44 × 10^−8^**
**BMI (kg/m^2^)**	22.42 ± 1.48 ****** ✦ **¶**	26.28 ± 5.03 ****^,^***	27.85 ± 5.10 ✦	29.41 ± 4.73 **¶ ***	**9.49 × 10^−7^**
**WHR**	0.77 ± 0.07 **† ¶** ✦	0.86 ± 0.12 **✦ ***	0.85 ± 0.08 **#**	0.90 ± 0.09 **† ¶ * #**	**2.74 × 10^−6^**
**AST (U/L)**	20.40 ± 4.89	21.44 ± 4.78	21.94 ± 6.48	21.22 ± 7.28	0.641
**ALT (U/L)**	20.27 ± 11.19	22.40 ± 9.39	25.06 ± 13.19	26.04 ± 12.89	0.272
**AST/ALT** **Ratio**	1.14 ± 0.36 †	1.06 ± 0.33	0.98 ± 0.29	0.90 ± 0.35 **†**	**0.005**
**GammaGT (U/L)**	18.80 ± 15.63	20.54 ± 15.35	23.36 ± 19.99	23.37 ± 15.78	0.643
**FGlu (mg/dL)**	85.27 ± 8.92	85.12 ± 7.93	86.01 ± 9.80	89.49 ± 12.03	0.063
**FIns (μU/mL)**	9.56 ± 3.35	11.18 ± 5.04	10.87 ± 4.75 **#**	14.14 ± 8.72 **#**	**0.003**
**HOMA-IR**	2.01 ± 0.88	2.38 ± 1.67 *****	2.05 ± 1.37 **#**	3.2 ± 1.88 *** #**	**0.003**
**TC (mg/dL)**	186.2 ± 45.51	203.14 ± 41.36	203.23 ± 35.86	204.04 ± 34.85	0.393
**LDL (mg/dL)**	108.66 ± 36.30	128.81 ± 37.09	126.39 ± 33.32	126.67 ± 26.29	0.194
**HDL (mg/dL)**	64.38 ± 16.15 **¶**	57.02 ± 12.39 *****	57.97 ± 14.05 **#**	49.76 ± 11.16 **¶ * #**	**2 × 10^−5^**
**TG (mg/dL)**	66.40 ± 45.30 **¶**	86.81 ± 43.38 *****	94.85 ± 54.06 **#**	117.85 ± 62.63 **¶ * #**	**0.001**
**Hyperlipidemia (%)**	26.6	58	50	56.5	0.149
**DMII (%)**	0	0	4	5	0.33
**Hypertension (%)**	13.3	34	32.9	55.2	**0.002**
**MetS (%)**	6	28	26.8	42.1	**0.024**

Values given as mean ± SD for quantitative variables and numbers or relative frequencies (%) for categorical variables. *p*-value: ANOVA *p*-value for quantitative and chi-square *p*-value for categorical variables. **†**, **¶**, **§**, *****, **#**, ******, ✦: *p* ≤ 0.05 for multiple comparisons using the Tukey’s post hoc test. *FLI*: Fatty Liver Index; *PAL*: Physical activity level; *BMI*: Body mass index; *WHR*: Waist-to-hip ratio; *AST*: Aspartate transaminase; *ALT*: Alanine transaminase; *GammaGT*: Gamma-glutamyltransferase; *FGlu*: Fasting glucose; *FIns*: Fasting insulin; HOMA-IR: Homeostasis Model Assessment—Insulin Resistance; TC: Total cholesterol; *LDL*: Low-density lipoprotein; *HDL*: High-density lipoprotein; *TG*: Triglycerides; *DMII*; Diabetes Mellitus II; *MetS*: Metabolic Syndrome.

## Data Availability

Data will be made available upon reasonable request due to their containing information that could compromise the privacy of research participants.

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
