# Peer review of "Circulatory Metabolite Ratios as Indicators of Lifestyle Risk Factors Based on a Greek NAFLD Case–Control Study"

_nutrients, 2024, doi:10.3390/nu16081235_

Round 1
Reviewer 1 Report
Comments and Suggestions for Authors
Overall, the manuscript has no great defects. However, personally, I think the novelty and significance are limited. Here are some minor concerns.
--line 77, should "[19][20]" be "[19-20]" or "[19,20]"?
--Why LS does not contain alcohol drinking? After all, drinking is also associated with MASLD besides alcohlic liver disease.
--Authors assess the stage of MASLD by ultrasound rather than biopsy, which may cause bias. I suggest authors discuss it.
--Line 133, only the data in normal distribution can be shown as "Mean±SD".
--The figures' resolutions seem to be low.
Author Response
We kindly thank reviewer #1 for his/her valuable comments to our work.
Overall, the manuscript has no great defects. However, personally, I think the novelty and significance are limited. Here are some minor concerns.
--line 77, should "[19][20]" be "[19-20]" or "[19,20]"?
We thank the reviewer for this remark. The suggested change was applied (line 76).
--Why LS does not contain alcohol drinking? After all, drinking is also associated with MASLD besides alcoholic liver disease.
In 2023, the new nomenclature of non-alcoholic fatty liver disease (NAFLD) was decided to be metabolic dysfunction-associated steatotic liver disease (MASLD). As mentioned in the section 2.1, excessive alcohol consumption is involved, among others, in the exclusion criteria for participant enrollment [1]. For this reason and since alcohol intake in this sample had low between-person variation, this parameter was not included in the construction of the LS.
--Authors assess the stage of MASLD by ultrasound rather than biopsy, which may cause bias. I suggest authors discuss it.
Thank you for your valuable comment. Indeed, liver biopsies are the diagnostic reference standard for MASLD, due to their high accuracy in detection and staging of the disease. However, biopsy per se has some disadvantages (it is invasive, expensive, and can occasionally result in serious side effects). On the other hand, ultrasound is commonly the first-line examination for individuals with a high risk of this disease, given that it is reasonably priced, widely accessible, non-invasive, suitable for follow-up exams, and incredibly practical for patients [2]. Taking into account your comment, the use of US has now been discussed in the section 2.1.1 (lines 96-107).
--Line 133, only the data in normal distribution can be shown as "Mean±SD".
We would like to thank the reviewer for the comment. We have now edited the manuscript accordingly (line 139).
--The figures' resolutions seem to be low.
We have made appropriate changes in the Figures
References
- Kalafati, I. P., D. Borsa, M. Dimitriou, K. Revenas, A. Kokkinos and G. V. Dedoussis. "Dietary patterns and non-alcoholic fatty liver disease in a greek case-control study." Nutrition 61 (2019): 105-10. 10.1016/j.nut.2018.10.032. http://www.ncbi.nlm.nih.gov/pubmed/30708259.
- Leoni, S., F. Tovoli, L. Napoli, I. Serio, S. Ferri and L. Bolondi. "Current guidelines for the management of non-alcoholic fatty liver disease: A systematic review with comparative analysis." World J Gastroenterol 24 (2018): 3361-73. 10.3748/wjg.v24.i30.3361. http://www.ncbi.nlm.nih.gov/pubmed/30122876.

Reviewer 2 Report
Comments and Suggestions for Authors
Here is a review report of the study "Circulatory metabolite ratios as indicators of lifestyle risk factor based on a Greek NAFLD case-control study":
The study investigates the association between circulatory metabolite ratios and lifestyle risk factors in Greek individuals with non-alcoholic fatty liver disease (NAFLD). It addresses an important gap in research by examining how lifestyle factors such as obesity, unhealthy diet, physical inactivity, and smoking influence the circulatory metabolome and are associated with the development of chronic diseases.
Some comments for improvement: please specify if the surveys were taken in person and if it was done by trained personnel.
Author Response
We kindly thank reviewer #2 for his/her kind comments to our work.
Here is a review report of the study "Circulatory metabolite ratios as indicators of lifestyle risk factor based on a Greek NAFLD case-control study":
The study investigates the association between circulatory metabolite ratios and lifestyle risk factors in Greek individuals with non-alcoholic fatty liver disease (NAFLD). It addresses an important gap in research by examining how lifestyle factors such as obesity, unhealthy diet, physical inactivity, and smoking influence the circulatory metabolome and are associated with the development of chronic diseases.
Some comments for improvement: please specify if the surveys were taken in person and if it was done by trained personnel.
Thank you for the comment. Taking into consideration your comment we have revised our manuscript accordingly (lines 109-110).

Round 2
Reviewer 1 Report
Comments and Suggestions for Authors
I think the manuscript can be accepted. I have no further questions but one. In line 139, author claims "all quantitative variables were normally distributed". In Table 1 [FLI], 8.34±7.36, 30.90±29.46. It is unlikely that the variables are normally distributed.
Author Response
We kindly thank reviewer #1 for his/her comments to our work.
I think the manuscript can be accepted. I have no further questions but one. In line 139, author claims "all quantitative variables were normally distributed". In Table 1 [FLI], 8.34±7.36, 30.90±29.46. It is unlikely that the variables are normally distributed.
We apologize for not fully clarifying our previous response regarding normality. Regardless of the distribution of the population, the central limit theorem (CLT) asserts that as sample sizes increase, the distribution of sample means approaches a normal distribution. Usually, sample sizes of 30 or more are thought to be adequate for the CLT to hold. Our sample size (n: 223) is much larger than 30, so we based on CLT and assumed that the variables are normally distributed. We have made appropriate changes to the manuscript (lines 139-140).